# Phenolic Plant Extracts Versus Penicillin G: In Vitro Susceptibility of *Staphylococcus aureus* Isolated from Bovine Mastitis

**DOI:** 10.3390/ph12030128

**Published:** 2019-08-31

**Authors:** Fernanda Gomes, Maria Elisa Rodrigues, Natália Martins, Isabel C.F.R. Ferreira, Mariana Henriques

**Affiliations:** 1CEB, Centre of Biological Engineering, LIBRO–Laboratório de Investigação em Biofilmes Rosário Oliveira, University of Minho, 4710-057 Braga, Portugal; 2Faculty of Medicine, University of Porto, Alameda Prof. Hernani Monteiro, 4200-319 Porto, Portugal; 3Institute for Research and Innovation in Health (i3S), University of Porto, Rua Alfredo Allen, 4200-135 Porto, Portugal; 4Centro de Investigação de Montanha (CIMO), Instituto Politécnico de Bragança, Campus de Santa Apolónia, 5300-253 Bragança, Portugal

**Keywords:** bovine mastitis, *Eucalyptus globulus* Labill., *Juglans regia* L., penicillin G, *Staphylococcus aureus*

## Abstract

Antibiotics are the elective drugs in bovine mastitis (BM) treatment, despite their low rates of efficiency and effectiveness and increasing risk of pathogen resistance. In this sense, it is urgent to discover new and effective antimicrobial agents to apply in BM control and even treatment. Plant extracts have been widely recognized as a rich source of phytochemicals with antimicrobial potential. Thus, the present work aims to compare the bioactivity of *Eucalyptus globulus* and *Juglans regia* extracts against *Staphylococcus aureus* bovine mastitis strains with penicillin G. At non-toxic concentrations, *E. globulus* exerted a bacteriostatic effect in planktonic cells and *J. regia* had no antimicrobial activity. Penicillin G, at minimum inhibitory concentration (MIC), demonstrated bactericidal activity, but just for *S. aureus* 3, 5, 6 and ATCC 25923, while the other strains seem to have acquired resistance. On the other hand, *E. globulus* and penicillin G in combination demonstrated synergy, being the most effective approach against *S. aureus* 1, 2 and 4. Thus, penicillin alone and in combination with *E. globulus* or *J. regia* seems to be promissory strategies to control bovine mastitis infections.

## 1. Introduction

Bovine mastitis (BM) is one of the most problematic infectious disorders with considerable economic losses for the dairy industry [1]. Antimicrobial agents, particularly antibiotics, have been widely used for the management and treatment of BM [2]. However, this kind of therapy has markedly promoted the emergence of antibiotic-resistant species, hampering mastitis management [1]. *Staphylococcus aureus (S. aureus)* is one of the most problematic causative agents of BM, being even recently highlighted by several studies that *S. aureus* contaminated milk from mastitis’ infected animals can enter in the dairy production chain and be the origin of food contamination [3,4,5]. Therefore, the search for natural and effective antimicrobial drugs for preventive and therapeutic purposes of BM is an urgent need. Natural matrices are rich sources of a great variety of useful bioactive molecules for the development of new antimicrobial products [6]. As an example, *Eucalyptus globulus* Labill., rich in volatile compounds, is a well-known medicinal plant due to its biological and pharmacological properties [7,8,9]. *E. globulus* essential oil has a great application in folk medicine for the treatment of several diseases, including mastitis [10]. On the other hand, leaves of *Juglans regia* L. are also widely used as antimicrobial in complementary and alternative therapies [11,12].

Nowadays, several research groups are testing fractions and even extracts from *J. regia* in animal models with different pathologies aiming to find potential therapeutic benefits [11]. Adnan et al. (2018) suggested that the antimicrobial activity of plants is primarily due to the presence of phenolic compounds [13], and therefore the higher the extraction of phenolic compounds the greater the antimicrobial efficacy. Moreover, although alcoholic solvents such as methanol and ethanol are preferentially used in plant extractions it was shown that phenolic extraction is significantly improved when a small portion of water is added to organic solvents [14]. 

Considering their promisor antimicrobial properties [15] and aiming to overcome the problem of antibiotic resistance, in this work the antibacterial activity of *Eucalyptus globulus* and *Juglans regia* methanol–water extracts were compared with penicillin G, alone and in combination, against *S. aureus* isolated from cows suffering mastitis. The evaluation of the cytotoxic potential of *E. globulus* and *J. regia* extracts were also performed.

## 2. Results

Table 1 shows the minimum inhibitory concentration (MIC) values for penicillin G. From the total BM isolates, three of them (i.e., *S. aureus* 1, 2 and 4 strains) presented an MIC value of 24 µg/mL, while the remaining ones, including *S. aureus* reference strain ATCC 25923, presented an MIC value of 0.4 µg/mL (Table 1). 

The MIC values obtained for penicillin G demonstrated that *S. aureus* 1, 2 and 4 strains present a similar level of susceptibility, while *S. aureus* 3, 5, 6 and *S. aureus* ATCC 25923 may be included in another group of susceptible strains. Therefore, based on their resistance patterns it was possible to divide the tested strains into two distinct groups, i.e., most resistant (*S. aureus* 1, 2 and 4) and less resistant (*S. aureus* ATCC 25923, 3, 5 and 6).

In order to know which concentrations of plant extracts to use in this study, the cytotoxicity of *E. globulus* and *J. regia* was evaluated. The percentage of viable animal cells obtained after exposure to both plant extracts was closely related to those of the negative control (cells grown without plant extracts), at concentrations ≤100 µg/mL of *E. globulus* (Figure 1a) and ≤195 µg/mL of *J. regia* (Figure 1b) extracts (*p* > 0.05). Specifically, for the *E. globulus* extract, at a concentration of 195 µg/mL, the percentage of viable cells (63.9%) was close to the limit to be considered non-cytotoxic (70%) based on ISO 10993-5:2006. Considering the results obtained, the antibacterial activity of *E. globulus* and *J. regia* (at non-cytotoxic concentrations) and penicillin G (at MIC) was tested alone and in combination against planktonic cells of *S. aureus*. Moreover, based on the MIC pattern of results, only a representative strain from each group was selected to be presented in Figure 2, namely *S. aureus* 1 and *S. aureus* ATCC 25923.

The use of *E. globulus* alone seems to exert a bacteriostatic effect against *S. aureus* until 8 h of incubation (Figure 2). Over time, *E. globulus* and *J. regia* alone presented a slight inhibitory effect, when compared with the positive control (Figure 2). Regarding penicillin G, no bioactivity was observed against *S. aureus* 1, 2 and 4 (Figure 2a), with only bactericidal effects being evident against all the other strains selected (Figure 2b). In this case, penicillin G promoted a log reduction ranging from 2.78 to 5.76, after 24 h (*p* < 0.05). On the other hand, when both extracts were combined with penicillin G, the most effective antimicrobial activity was observed for *S. aureus* 1, 2 and 4 strains, compared to the antimicrobial agent tested alone (Figure 2a).

## 3. Discussion

Penicillin is the most commonly used drug in the treatment of BM, even though their use is closely related to the increasing appearance of resistant strains [16]. In this sense, the assessment of the feasibility of using a combinatory strategy might provide considerable benefits, even allowing a greater efficacy (synergistic or potentiation reaction) at lower doses, reducing the development of drug-resistant species and even preventing the occurrence of side effects [17]. Thus, the evaluation of the antimicrobial activity of both plant extracts against planktonic cells of *S. aureus* BM isolates and *S. aureus* ATCC 25923 were performed under different conditions, which included: The use of *E. globulus* and *J. regia* alone and in combination, at non-cytotoxic concentrations; and the use of penicillin G at MIC value, alone and in combination with both plant extracts. 

Then, the assessment of the cytotoxic potential and safety of both extracts was performed, and the results obtained for both matrices revealed that *E. globulus* and *J. regia* are non-toxic to animal cells at concentrations ≤100 and ≤195 µg/mL, respectively (Figure 1a,b). When both extracts were used at non-cytotoxic concentrations and compared with the antibacterial effect exerted by penicillin G, the most prominent effect was observed to *E. globulus* extract against *S. aureus* 1, 2 and 4 strains (Figure 2a). For those strains, *E. globulus* evidenced a bacteriostatic effect until 8 hours of incubation. On the other hand, no antimicrobial effects were observed with penicillin G. In fact, although *E. globulus* exerted a similar bacteriostatic effect against the remaining strains, penicillin G was the most effective agent, even exerting bactericidal effects (Figure 2b). Moreover, the obtained penicillin G MIC value to *S. aureus* 1, 2 and 4 strains was nearly 60 times higher than those obtained with *S. aureus* 3, 5, 6 and ATCC 25923 (Table 1), being, therefore, more resistant to penicillin G.

Synergistic effects were observed to *S. aureus* 1, 2 and 4 strains to the combinations *E. globulus* + penicillin G and *J. regia* + penicillin G being first the most effective combination. However, the combination of these agents with *J. regia* did not further improve their antimicrobial activity. For these strains, plant extracts combined with penicillin G seemed to exert an immediate antimicrobial effect and the number of colony forming units (CFUs) decreased dramatically up to 8 h. However, CFUs increased after 24 h incubation under the combination strategy (Figure 2a). Perhaps a new dose of plant extracts will necessarily be added after 6/8 h of treatment in order to obtain a higher and prolonged inhibitory activity. Concerning the *S. aureus* ATCC 25923, 3, 5 and 6 strains, penicillin G used alone still continued to be more effective than all the combinations tested. Moreover, penicillin G in combination with plant extracts promoted, over time and up to 24 h of incubation, an increasing CFU log reduction probably due to the bactericidal effect of penicillin G (Figure 2b). 

In general, this study revealed an interesting synergy of *E. globulus* or *J. regia* in combination with penicillin G for most penicillin-resistant strains. Penicillin G is a β-lactam antibiotic that inhibits cell wall synthesis. On the other side, the major phenolic compounds found in *E. globulus* extract belong to the phenolic acids (gallic acid, caffeic acid and ellagic acid) and the flavonoids (quercetin) group [15]. Specifically, the most common phenolic compounds found in *E. globulus* and *J. regia* extracts are caffeic acid derivatives and quercetin. When looking at the specific mechanisms of action, gallic and caffeic acids mainly inhibit the H^+^-ATPase, responsible for ATP production [18] The ellagic acid interacts with enzymes, consequently inhibiting their interaction with proteins [18]. Flavonoids affect the cell wall stability, promoting the disruption of cell structure [18]. Thus, the major phenolic compounds found in *E. globulus* and *J. regia*, gallic acid and quercetin derivatives, respectively, demonstrated a great ability to strengthen the bioactivity of β-lactam antibiotics by inhibiting the penicillin binding protein 2a (PBP2a) [19]. This peptidoglycan transpeptidase, in cooperation with the transglucosylase domain of PBP2 of *S. aureus,* is responsible for bacterial proliferation when exposed to β-lactam antibiotics [20]. Therefore, PBP2a inhibitors are usually used to overcome the antibiotics resistance and it is probably the mechanism responsible for the synergy observed when antimicrobials combination was tested. 

It is also interesting to highlight that, to the most susceptible strains, penicillin G is still the most effective approach. Moreover, it should be also pointed out that, in vitro studies might not reflect what really occurs in vivo. In fact, during administration, natural molecules suffer numerous biochemical reactions in organisms while some others need to be metabolized to become biologically active, and other become inactive [21]. Furthermore, the in vivo bioavailability of antimicrobial drugs may vary both due to endogenous factors and even inter and intra-individual variations.

## 4. Materials and Methods

### 4.1. Samples

In this study, the leaves of two plant species were used: *Eucalyptus globulus* Labill. (blue gum) from commercial origin (Cantinho das Aromáticas, an organic and certified farm from Vila Nova de Gaia, Portugal), and *Juglans regia* L. (walnut) harvested in Trás-os-Montes, Bragança, Portugal, collected and identified according to Santos et al. [12].

### 4.2. Standards and Reagents

Methanol was of analytical grade purity and supplied by Pronalab (Lisbon, Portugal). Tryptic Soy Broth (TSB) medium and Tryptic Soy Agar (TSA) were purchased from Liofilchem (Roseto degli Abruzzi, Italy) and Merck (Darmstadt, Germany), respectively, and prepared according to the manufacturer’s instructions. Water was treated in a Milli-Q water purification system (TGI Pure Water Systems, Greenville, SC, USA). Penicillin G was purchased from Sigma Chemical Co., (St. Louis, MO, USA).

### 4.3. Preparation of the Extracts

Methanol–water extracts were obtained by extracting each plant sample (1 g) with 30 mL of methanol–water (80:20, *v*/*v*) at 25 °C and 150 rpm for 1 h, and filtering through Whatman No. 4 paper. The final residue was then extracted with an additional 30 mL portion of methanol–water mixture. Each one of the combined extracts was evaporated at 35 °C under reduced pressure (rotary evaporator Büchi R-210, Flawil, Switzerland) and then lyophilized (FreeZone 4.5, Labconco, Kansas City, MO, USA). The lyophilized methanol–water extracts were re-dissolved in water to obtain stock solutions at 50 × 10^3^ µg/mL, from which several dilutions were prepared. *E. globulus* extract was previously characterized in terms of chemical compounds [12,15,22], and *J. regia* (methanolic and aqueous extracts) was previously characterized by Santos et al. [12]. 

### 4.4. Determination of Minimal Inhibitory Concentrations (MICs)

Seven *S. aureus* were selected for this work: Reference strain (*S. aureus* ATCC 25923) and six BM isolates, kindly provided by the Portuguese lab (Segalab-Laboratório de Sanidade Animal e Segurança Alimentar SA). Minimal inhibitory concentration (MIC) of penicillin G was determined by the microbroth dilution technique. The determination of MIC values was performed by the serial two-fold dilutions method at concentrations ranging from 2 × 10^−2^ µg/mL to 24 µg/mL. The final cell concentration used was 5 × 10^5^ cells/mL. Then, 96-well plates (Orange Scientific, Braine-l’Alleud, Belgium) were incubated at 37 °C for 24 h. Samples (cells grown with penicillin G) and bacteria-free controls (positive controls) were also included. After visualization of the resultant plate, MIC values were correspondent to the antibiotic concentration where there was no visible growth, or even bacteriostatic effect by comparison with the positive controls. The number of viable cells was assessed through determination of a number of colonies forming units (CFUs), after 24 h of incubation at 37 °C. The results were presented as total of CFUs (Log CFUs). All the experiments were carried out in triplicate and repeated at three different moments.

### 4.5. Cytotoxicity Assay

The effect of *E. globulus* and *J. regia* on cell viability was assessed by the MTS ((3-[4,5-carboxymethoxyphenyl]-2-(4-sulfophenyl)-2H-tetrazolium), Promega) assay, using human primary fibroblast cell lines (3T3-CCL 163-from the American Type Culture Collection). Briefly, the cells were trypsinized and seeded in 96-well plates at a final concentration of 1 × 10^5^ cells/mL, in Dulbecco Modified Eagle Medium (DMEM, Biochrom GmbH) supplemented with 10% of fetal bovine serum (FBS, Sigma Aldrich) and 1% of penicillin/streptomycin (Biochrom AG), at 37 °C and 5% CO_2_. After 24 h of incubation, the cell cultures were treated with several concentrations of *E. globulus* and *J. regia* dissolved in DMEM complete medium, for a further 24 h. After this time, the wells were washed twice with phosphate buffered saline (PBS) (1×), and then 150 µL of MTS with DMEM without phenol (1/10, *v*/*v*) (Sigma Aldrich) were added to each well. After 1 h, the cell viability was assessed by reading the absorbance at 490 nm (absorbance of the orange formazan product) in a microplate reader Sunrise (Männdorf, Switzerland). The results were expressed as a % of viable cells, corresponding the OD_490_ to cells grown without extracts as 100% of cell viability. Dimethyl sulfoxide (DMSO, Sigma Aldrich) at 50% (*v*/*v*) in DMEM was used as positive control. 

### 4.6. S. aureus Susceptibility to Eucalyptus globulus, Juglans regia and Penicillin G

The antibacterial activity of *E. globulus* (100 µg/mL) and *J. regia* (195 µg/mL) in planktonic cells was tested against seven different *S. aureus* strains: One reference strain (*S. aureus* ATCC 25923) and six clinical isolates (*S. aureus* 1, 2, 3, 4, 5 and 6). For that, strains (1 × 10^6^ cells/mL) were grown in Erlenmeyer flasks in the presence of each selected plant extract, alone and in combination. Then, the cells were incubated at 37 °C and 120 rpm for 24 h. The surviving viable bacteria were obtained by CFU assay after 0, 2, 4, 6, 8 and 24 h of incubation. The antimicrobial activity of selected plant extracts was compared with the effect of penicillin G at determined MIC concentration, following the same procedure. Antibacterial activity of penicillin G alone and in combination with *E. globulus* and *J. regia* was also tested. A synergistic effect was considered when the bioactivity of the antimicrobials in combination was higher than the sum of the effect of each agent alone [23]. All experiments were performed in triplicate and repeated on three different occasions. 

### 4.7. Statistical Analysis

Data were analyzed using one-way analysis of variance (ANOVA) and means were compared using Tukey’s honestly significant difference (HSD) multiple comparisons test. All statistical tests were performed using the Prism software package (GraphPad Software version 6.0 for Macintosh). Results were considered statistically significant when *p* < 0.05.

## 5. Conclusions

In overall, this study highlighted the problem of emerging resistant species to antimicrobial agents. In fact, penicillin G, one of the main therapeutic strategies used in the prevention and treatment of BM, seemed to be ineffective against planktonic cells of the most resistant strains (50% of the tested isolates). On the other hand, non-cytotoxic concentrations of *E. globulus* exerted a bacteriostatic effect. Moreover, depending on the level of resistance evidenced by *S. aureus* strains to penicillin G, penicillin alone and in combination with *E. globulus* and/or *J. regia* seems to be promissory strategies to control bovine mastitis infections. In this sense, this study drew attention to a worrying problem concerning BM and evidenced the clear and urgent need to discover effective therapeutic strategies to control and to manage this pathology. 

## Figures and Tables

**Figure 1 pharmaceuticals-12-00128-f001:**
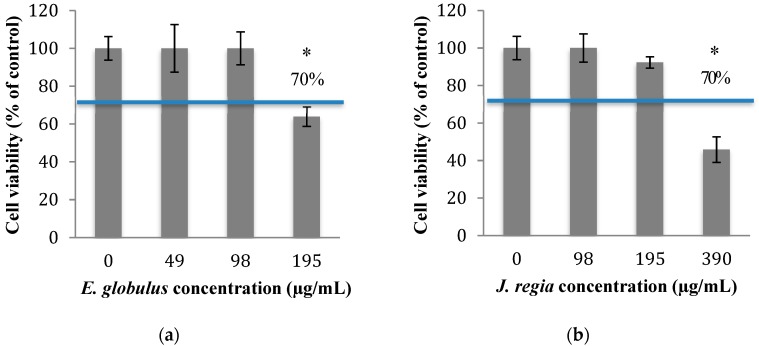
Fibroblast 3T3 viability after 24 h of contact with different concentrations of *E. globulus* (**a**) and *J. regia* (**b**) extracts, measured with MTS assay. The negative control allowed the perfect growth of the cells (cells grown without extracts; 100% viability). * *p* < 0.05.

**Figure 2 pharmaceuticals-12-00128-f002:**
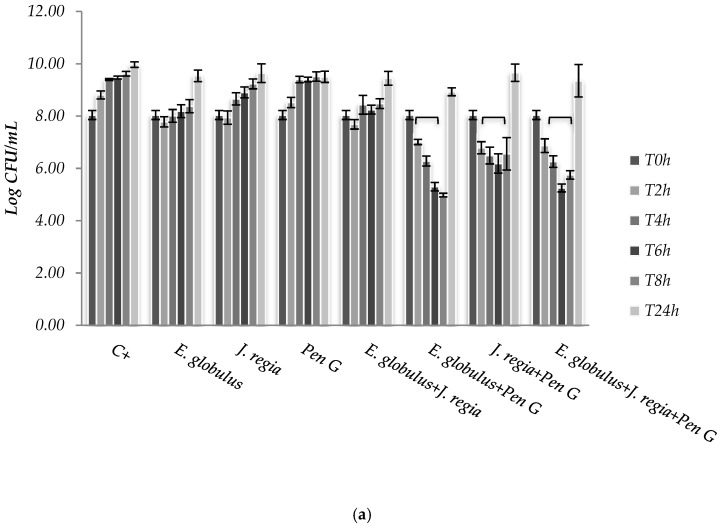
Colony forming units (CFUs) of *Staphylococcus aureus* 1 (**a**) and ATCC 25923 (**b**), cultured within methanol–water extract of *E. globulus* (100 µg/mL), *J. regia* (195 µg/mL) and penicillin G (MIC), alone and in combination. C+ - Positive control. * *p* < 0.05.

**Table 1 pharmaceuticals-12-00128-t001:** Minimum inhibitory concentration (MIC) of penicillin G against *S. aureus* isolates (*n* = 7), determined by the broth microdilution method.

MIC (µg/mL)							
Strains	*S. aureus* 1	*S. aureus* 2	*S. aureus* 3	*S. aureus* 4	*S. aureus* 5	*S. aureus* 6	*S. aureus* ATCC 25923
Pen G	24	24	0.4	24	0.4	0.4	0.4

Pen G: Penicillin G.

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
