# Peer review of "Phenolic Plant Extracts Versus Penicillin G: In Vitro Susceptibility of Staphylococcus aureus Isolated from Bovine Mastitis"

_pharmaceuticals, 2019, doi:10.3390/ph12030128_

Round 1

Reviewer 1 Report

According to the authors, the manuscript is planned to be published as a short communication, however, on the draft it says "article". As mentioned in my first review, I think the amount of new information presented is not sufficient for a full length article, but if it is submitted as short communication, I would accept it.

Author Response

The reviewer suggestion was carefully considered. The article type was changed to short communication

Reviewer 2 Report

Dear Editor and Authors,

I carefully read the manuscript “Phenolic plant extracts versus penicillin G: in vitro susceptibility of Staphylococcus aureus isolated from bovine mastitis” by Fernanda Gomes et al. The study is interesting and the manuscript is well organized. However the quality and importance of both can be improved.

General comments:

Introduction: Better to add few sentences about antimicrobial resistance in S. aureus to highlight the importance of this study. Missing references: lines 34, 35, 46.

Results: the choice of authors to present the bacteriostatic values for only two representative strains limit the importance and value of this study. I suggest adding a table with all the values. Moreover the MIC seems to be determined only for Penicillin, I think is important to be determined and added also for extracts even if the authors used the cytotoxicity level.

From my point of view the importance of the study will increase considerably if the authors decide to determine the phenolic compounds content since this may vary depending on various factors.

Discussion: are other studies on this two extracts on S. aureus? If yes should be added and discussed.

Specific comments:

Line 45 including mastitis

Line 49 is primarily due to… this sentence as it is written suggest that only phenolic compounds have antimicrobial activity but in the absence of strong papers to demonstrate this other compounds can’t be excluded. Better to rephrase this sentence entirely

Author Response

Introduction: Better to add few sentences about antimicrobial resistance in S. aureus to highlight the importance of this study. Missing references: lines 34, 35, 46.

Answer: The antimicrobial resistance is mentioned in the text as a general problem of all bovine mastitis causative agents including S. aureus as one of the most problematic causative agents. “antibiotics…markedly promoted the emergence of antibiotic-resistant species, hampering mastitis management [1]. Staphylococcus aureus is one of the most problematic causative agents of BM…”

The missing references were introduced in the respective sentences.

Results: the choice of authors to present the bacteriostatic values for only two representative strains limit the importance and value of this study. I suggest adding a table with all the values.

Answer: The authors thank the reviewer’s suggestion but in our opinion a table with all values will not bring anything new to the article. It is mentioned and justified in the text the choice of the strains and aimed to simplify the presentation of results. On a graph it is easier to visualize the results obtained.

Moreover, the MIC seems to be determined only for Penicillin, I think is important to be determined and added also for extracts even if the authors used the cytotoxicity level.

Answer: The MIC values for both plant extracts studied in this paper were determined previously and published in another paper (Gomes et al., 2018- Ref 13)

Line 54 “Considering their promisor antimicrobial properties [13]…”

From my point of view the importance of the study will increase considerably if the authors decide to determine the phenolic compounds content since this may vary depending on various factors.

Answer: The authors agree with the reviewer. The phenolic characterization of the hydromethanolic extract of E. globuluswas already performed and can be found in the paper Gomes et al., 2018 (ref 13). The reference was wrong in the paper but was corrected.

Discussion: are other studies on these two extracts on S. aureus? If yes should be added and discussed.

Answer: There are studies of antibacterial activity of these two plants on S. aureus, however to the authors knowledge no studies are available using hydroalcoholic extracts and specifically hydromethanolic extracts.

Specific comments:

Line 45 including mastitis

Answer: This sentence was corrected as suggested by the reviewer.

Line 49 is primarily due to… this sentence as it is written suggest that only phenolic compounds have antimicrobial activity but in the absence of strong papers to demonstrate these other compounds can’t be excluded. Better to rephrase this sentence entirely

Answer: The sentence was rephrased as suggested by the reviewer.

Round 2

Reviewer 2 Report

Dear editor and authors,

The authors address all my comments therefore I consider the manuscript suitable for publication.

This manuscript is a resubmission of an earlier submission. The following is a list of the peer review reports and author responses from that submission.

Round 1

Reviewer 1 Report

Thank you for the revised manuscript. I was not aware that it was planned to be handed in as a short communication which fits much better for the presented data than a full length article. There are just a few minor issues from my side:
•    You mentioned that “our objective was to assess the antimicrobial effect of hydroethanolic extracts rich in phenolic compounds.” You could add this to the introduction section then the choice of the extraction solvent becomes clearer to the reader.
•    In line 106-108 of the discussion part you mention that the MIC for penicillin was determined for all strains and you’ve therefore grouped them into two groups from which you used 1 representative strain (1 and ATCC25923). It would be easier to follow this explanation if you added the information in the results part, because it is a bit confusing to only see two strains  in figure 2 while in the text you mention all strains referring to figure 2a (line 78/79).
•    Figure 2: it would be good to explain “C+” in the legend.
•    Line 123: the sentence is not well understandable, something like ”[…] however, the combination of these agents with J. regia did not further improve the activity.” Would be better
•    Please add the information about the origin of the isolates in the Materials section. Even if they were not further characterized, I think it’s better to include the information as it is than not mentioning it at all.

Reviewer 2 Report

The current manuscript reports the synergetic effect of plant extracts and antibiotic combinations against S. aureus. Assay results are clear and manuscript prepared well. However, there are some points require the attention of the authors as below;

1. There is no direct evidence suggested that the plant extracts are phenolic or not. This is one of the main ideas of this manuscript which was highlighted in the title.

2. There are many references about the antibacterial activity of E. globulus and J. regia against S. aureus. It is interesting that combination of the extracts shows stronger activities, however, this is not surprising for most of the readers. Also, figure 2 shows the opposit effect of combination that needs to be explained in detail.

3. Authors suggest that this manuscript revealed an interesting synergy of plant extracts combination. However, there are several similar experiments such as African J Basic & Appl Sci 2010 2 25 / Exp Bio Med 2017 242 731 / Asian Pacific J Trop Med 20103, 266 / African J. Biotech 2006 5 1082

It is not easy to find novel aspects of current manuscript compare with the papers above.

Reviewer 3 Report

This short study shows the synergy of extracts of J regis and E globulus against S aureus strains. A major issue I have with this is that I am not convinced it shows anything new at all. I am rather surprised that the authors don't cite, for instance,  Farooqui et al (2015) PlosOne 10(2):e011843. This is That study is far more comprehensive than what was presented here, which included synergy of J regia extracts with several antibiotics against 350 bacteria. 

Specific comments:

1. The text claims that figure 2 shows for multiple strains, but the results for only 1 strain is shown, without knowing that these data are indeed representative. Data from all strains should be shown. 

2. If the authors claim synergy, they should also perform checkerboard assays, which is the gold standard for determining synergy/antagonism.

3. The discussion mentions that penG alone was better for the sensitive strain, but it should have been more clearly stated that there actually appears to be antagonism. Again, to make a proper judgement on synergy/antagonism, checkerboard assays should be performed which allows the calculation of the fractional inhibitory concentration index.

4. What is the origin of the S. aureus strains 1-6? This should be described. Have they been published before? From where were they collected? How was the species confirmed? etc.

5. The MIC tests were incubated for 24-48 hours. The outcome can be actually quite different whether these tests are incubated for 24 or 48 hours, so to just incubate for anything between 24 and 48 hours is not acceptable. Also, for standardised methods (as recommended by eg CLSI) a medium such as Mueller Hinton broth should be used, not TSB.